# TOWARDS REPLICATION-ROBUST DATA MARKETS

## ABSTRACT

Despite widespread adoption of machine learning throughout industry, many firms face a common challenge: relevant datasets are typically distributed amongst market competitors that are reluctant to share information. Recent works propose *data markets* to provide monetary incentives for *collaborative* machine learning, where agents share features with each other and are rewarded based on their contribution to improving the predictions others. These contributions are determined by their relative *Shapley value*, which is computed by treating features as players and their interactions as a characteristic function game. However, in its standard form, this setup further provides an incentive for agents to replicate their data and act under multiple false identities in order to increase their own revenue and diminish that of others, restricting their use in practice. In this work, we develop a *replication-robust* data market for supervised learning problems. We adopt Pearl's *do*-calculus from causal reasoning to refine the characteristic function game by differentiating between observational and interventional conditional probabilities. By doing this, we derive Shapley value-based rewards that are robust to this malicious replication by design, whilst preserving desirable market properties.

## 1 INTRODUCTION

When faced with machine learning task, it can often be the case that a firm would benefit from using the data of others. For example, rival distributors of similar goods may improve supply forecasts by sharing sales data, hoteliers could find value in data from airline companies for anticipating demand, hospitals could reduce social biases from diagnostic support systems by sharing patient details, and so forth. In this work, we consider the example of renewable energy producers exposed to uncertain levels of production and therefore require reliable forecasts to competitively participate in electricity markets, with their revenue a function of predictive performance. It is well-studied that, with access to distributed data, in both a geographic and ownership sense, these agents could exploit spatial and temporal correlations between sites to improve their forecasts (Tastu et al., 2013).

In practice, firms may be reluctant to share information due to privacy concerns or perceived conflicts of interest. Whilst methods from the field of federated learning (Lalitha et al., 2018) could indeed be used to train models on local servers without the need to centralize any data, this relies on altruistic sharing of information amongst market competitors. An alternative approach is to provide incentives for data sharing—recent works propose *data markets* (Bergemann & Bonatti, 2019), where agents can collaborate by sharing features with each other to improve the predictions of others, without transferring any raw data between them (Pinson et al., 2022). With foundations in informational efficiency of financial markets (Hayek, 1986), data markets have similar economic roots as *prediction markets* (Waggoner et al., 2015), mechanisms designed to consolidate information with the goal of forecasting outcomes of future events (Frongillo & Waggoner, 2018).

Whilst prediction markets can also be used to crowdsource data for machine learning (Abernethy & Frongillo, 2011), data owners themselves need to decide which tasks to contribute to *a priori*, when the relevance of their dataset is unknown. In contrast, data markets serve as real-time mechanism that match features to machine learning tasks based their capacity to improve predictive performance. Market revenue is a function of the value this brings to the task owner, and each feature owned by a distributed agent is rewarded based on its marginal contribution to the improvement.

**Challenges** Marginal contributions are hard to quantify when features are correlated. For example, if features are valued sequentially, it has been shown that agents would eventually sell their data for

less than their own costs as the information becomes redundant (Acemoglu et al., 2022). Whilst this is not the case if valuation occurs in parallel, the value of overlapping information is inherently combinatorial. To address this issue, recent works propose to adopt concepts from cooperative game theory, treating features as players and their interactions as a characteristic function game (Ghorbani & Zou, 2019). For many practitioners, the Shapley value (Shapley, 1997) is the solution concept of choice for such a game, which allocates each player its expected marginal contribution towards a set (or coalition) of other players, satisfying a collection of axioms that yield several desirable market properties by design (Agarwal et al., 2019). However, when computing this expectation, these works make implicit assumptions about the distribution of *in-* and *out-of-coalition* feautres, which creates incentives that result in grossly undesirable outcomes.

Specifically, these works model *observational* conditional probabilities for *out-of-coalition* features, which we show allows agents to replicate their data and act under multiple false identities to increase their revenue. For instance, if an agent's feature is highly correlated with that of another agent, they can simply submit many replicates of their feature under different identities, increasing their overall revenue and driving that of the other agents to zero. This stems from the fact that data, unlike material commodities, can be replicated at no additional cost. Whilst several attempts have been made to remedy this problem, doing so typically requires a trade-off. For instance, Ohrimenko et al. (2019) propose a more elaborate mechanism design, necessitating that each seller also owns a machine learning task, which has practical limitations. Agarwal et al. (2019) propose a modification to the Shapley value which penalizes similar features, thereby preventing replication, yet foregoes budget balance and remains vulnerable to spiteful agents—those who seek to minimize the revenue of other agents whilst maximizing their own profits. A similar shortcoming is observed in the proposal of Han et al. (2023), as both natural correlations and deliberate replications are penalized.

**Contributions**   The main contributions of our paper are as follows: (i) we propose a general data market design for supervised learning problems that subsumes many existing proposals in literature; (ii) we show that there are many ways in which Shapley values can be used to derive rewards and that the differences between them can be explained from a caused perspective; (iii) we leverage Pearl's seminal work on causality (Pearl, 2012) to show that by replacing the conventional approach of conditioning by *observation* with conditioning by *intervention*, we can design a data market in a way that is *replication-robust* whilst also accounting for spiteful agents, thereby taking a step toward practical application of these markets; finally (iv) we demonstrate our findings using a real-world case study—out of many potential applications, we choose to study wind power forecasting due to data availability, the known value of sharing distributed data, and the fact it is a sandbox that can be easily shared and used by others.

The remainder of this paper is structured as follows: Section 2 presents our general market design framework. In Section 3 we derive variants of the characteristic function and analyse each from a causal perspective. In Section 4 we discuss the implications of these to replication-robustness of the market. Section 5 then illustrates our findings on a real-world case study. Finally, Section 6 gathers a set of conclusions and perspectives for future work.

## 2   PRELIMINARIES

Throughout our work, we consider regression models to be used for forecasting, however our setup can readily be extended to general supervised learning problems. We build upon prior work on data acquisition for machine learning tasks from both strategic (Dekel et al., 2010) and privacy-conscious (Cummings et al., 2015) agents. In particular, we characterize an owner of a regression task by their valuation for a marginal improvement in predictive performance, which sets the price of the data of the distributed agents, whom in turn propose their own data as features and are eventually rewarded based on their relative marginal contributions. We denote this valuation $\lambda \in \mathbb{R}_{\geq 0}$, the value of which we assume to be known. The reader is referred to Ravindranath et al. (2024) for a recent proposal of how $\lambda$ may be learnt in practice.

**Market Agents**   The set $\mathcal{A}$ is the market agents, one of which $c \in \mathcal{A}$ is a *central agent* seeking to improve their predictions, whilst the remaining agents $a \in \mathcal{A}_{-c}$ are *support agents*, whom propose their own data as features, whereby $\mathcal{A}_{-c} = \mathcal{A} \setminus \{c\}$. Let $y_t \in \mathbb{R}_+$ be the target signal recorded by the central agent at time $t$, a sample from the stochastic process $\{Y_t\}_{\forall t}$. We write $\boldsymbol{x}_{\mathcal{I},t}$ as the vector

of all features at time $t$, indexed by the ordered set $\mathcal{I}$. Each agent $a \in \mathcal{A}$ owns a subset $\mathcal{I}_a \subseteq \mathcal{I}$ of indices. For each subset of features $\mathcal{C} \subseteq \mathcal{I}$ we write $\mathcal{D}_{\mathcal{C},t} = \{\boldsymbol{x}_{\mathcal{C},t'}, y_{t'}\}_{\forall t' \leq t}$ to be the set of observations up until time $t$.

**Regression Framework**  To model the target signal, $Y_t$, we use a parametric Bayesian regression framework, formulating the likelihood as a deviation from a deterministic mapping under an independent Gaussian noise process, the variance of which is treated as a hyperparameter. The mapping, $f$, is a linear interpolant parameterized by a vector of coefficients, $\boldsymbol{w}$, and represents the conditional expectation of the target signal, such that the interpolant corresponding to the *grand coalition* (i.e., using all available input features) at any particular time step can be decomposed as follows:

$$f(\boldsymbol{x}_t, \boldsymbol{w}) = w_0 + \underbrace{\sum_{i \in |\mathcal{I}_c|} w_i x_{i,t}}_{\substack{\text{Terms belonging} \\ \text{to the central agent.}}} + \underbrace{\sum_{a \in \mathcal{A}_{-c}} \sum_{j \in |\mathcal{I}_a|} w_j x_{j,t}}_{\substack{\text{Terms belonging} \\ \text{to the support agents.}}} . \tag{1}$$

**Market Clearing**  As in Pinson et al. (2022), we consider a two-stage (i.e., in-sample and out-of-sample) batch market, but relax the assumption that features are independent, yet still assume that any redundant features owned by the support agents (i.e., those that are highly correlated with the central agent's features) are removed via the detailed feature selection process. An important step in the market clearing procedure is parameter inference—to mitigate bias we opt for a centred isotropic Gaussian prior, which is conjugate for our likelihood, resulting in a tractable Gaussian posterior that summarizes our updated beliefs, which, for a particular subset of features is given by

$$p(\boldsymbol{w}_\mathcal{C} | \mathcal{D}_{\mathcal{C},t}) \propto p(\mathcal{D}_{\mathcal{C},t} | \boldsymbol{w}_\mathcal{C}) p(\boldsymbol{w}_\mathcal{C} | \mathcal{D}_{\mathcal{C},t-1}), \quad \forall t,$$

where recall $\mathcal{D}_{\mathcal{C},t}$ is the set of input-output pairs observed up until time $t$. We note that adoption of a Gaussian framework is merely for mathematical convenience, and our framework can be readily extended to more general hypotheses. Market revenue is a function of the exogenous valuation, $\lambda$, and the extent to which model-fitting is improved, which we measure using the negative logarithm of the predictive density (i.e., the convolution of the likelihood with the posterior), denoted by $\ell_t = -\log[p(y_t | \boldsymbol{x}_t)]$, $\forall t$, such that for a batch of observations, the market revenue is $\pi = \lambda(\mathbb{E}[\ell_t]_{\mathcal{I}_c} - \mathbb{E}[\ell_t]_{\mathcal{I}})$, which equals the payment collected from the central agent.

**Revenue Allocation**  To allocate market revenue amongst support agents, we define an attribution policy based on the Shapley value. We let $v : \mathcal{C} \in \mathcal{P}(\mathcal{I}) \mapsto \mathbb{R}$ be a characteristic function that maps the power set of indices of all the features to a real-valued scalar—the set $\mathcal{C}$ represents a coalition in the cooperative game. If we let $\Theta$ be the set of all possible permutation of indices in $\mathcal{I}_{-c}$, the Shapley value is $\phi_i = 1/|\mathcal{I}_{-c}|! \sum_{\theta \in \Theta} \Delta_i(\theta)$, $\forall i \in \mathcal{I}_{-c}$, where $\Delta_i(\theta) = v(\mathcal{I}_c \cup \{j : j \prec_\theta i\}) - v(\mathcal{I}_c \cup \{j : j \preceq_\theta i\})$, where $j \prec_\theta i$ means $j$ precedes $i$ in permutation $\theta$. The reward for each support agent can be written as $\pi_a = \sum_{i \in \mathcal{I}_a} \lambda \mathbb{E}[\phi_i]$, $\forall a \in \mathcal{A}_{-c}$. Therefore, all of the revenue is contained within the market, that is, $\pi = \sum_{a \in \mathcal{A}_{-c}} \pi_a$, and hence budget balance is attained.

We acknowledge that this formulation of the Shapley value endures a time complexity of $\mathcal{O}(2^{|\mathcal{I}_{-c}|})$, hence in practice one must rely on approximation methods (Castro et al., 2009; Mitchell et al., 2022; Zhang et al., 2023). For instance, we can take a uniform sample of permutations, $\mathcal{P} \subset \Theta$, and then compute a Monte Carlo estimate which, by the Central Limit Theorem, converges asymptotically at a rate of $\mathcal{O}(1/\sqrt{\mathcal{P}})$. Still though, evaluating the loss function using each subset of features is not that straightforward in general—once trained, machine learning models typically require an input vector containing a value for each feature to avoid matrix dimension mismatch. Hence, the characteristic function must *lift* the original loss to simulate removal of features (Merrill et al., 2019).

Recall that our loss function, $\ell$, relates to the mapping $f : \mathbb{R}^{|\mathcal{I}|} \mapsto \mathbb{R}$ described in (1), and is therefore itself only defined on $\mathbb{R}^{|\mathcal{I}|}$. To calculate the Shapley value, a value for each of the $2^{|\mathcal{I}|}$ subsets of input features is needed. Accordingly, we lift the loss function to the space of all subsets of features by formulating the characteristic function mapping as $v(\mathcal{C}) : \mathbb{R}^{|\mathcal{I}|} \times 2^{|\mathcal{I}|} \mapsto \mathbb{R}$, $\forall \mathcal{C}$. For the grand coalition, $v(\mathcal{I}) = \mathbb{E}[\ell_{\mathcal{I},t} | \boldsymbol{x}_t]$. The Shapley value is hence not well-defined in general, as there exists many methods to formulate the lift, and hence the characteristic function itself (Sundararajan & Najmi, 2020). In the following section, we shall explore these possible methods and their differences from a causal perspective.

## 3  CHARACTERISTIC FUNCTION

Commonly adopted lifts can broadly be categorized as either *observational* or *interventional*, which affect the characteristic function that underpins the cooperative game. The former is typically found in work related to data markets (e.g, Agarwal et al., 2019; Pinson et al., 2022). The observational lift uses the *observational conditional expectation*, the expectation of the loss over the conditional density of out-of-coalition features, given in-coalition take on their observed values, such that

$$v^{\mathrm{obs}}(\mathcal{C}) = \int \mathbb{E}\left[\ell_t | \boldsymbol{x}_{\mathcal{C},t}, \boldsymbol{x}_{\overline{\mathcal{C}},t}\right] p(\boldsymbol{x}_{\overline{\mathcal{C}},t} | \boldsymbol{x}_{\mathcal{C},t}) d\boldsymbol{x}_{\overline{\mathcal{C}},t}, \tag{2}$$

where $\overline{\mathcal{C}} = \mathcal{I} \setminus \mathcal{C}$ denotes the out-of-coalition features.

We propose to instead use the interventional lift, which uses the *interventional conditional expectation*, where features in the coalition are manually fixed to their observed values to manipulate the data generating process, expressed mathematically using Pearl's *do*-calculus (Pearl, 2012), such that

$$v^{\mathrm{int}}(\mathcal{C}) = \int \mathbb{E}\left[\ell_t | \boldsymbol{x}_{\mathcal{C},t}, \boldsymbol{x}_{\overline{\mathcal{C}},t}\right] p(\boldsymbol{x}_{\overline{\mathcal{C}},t} | do(\boldsymbol{x}_{\mathcal{C},t})) d\boldsymbol{x}_{\overline{\mathcal{C}},t}. \tag{3}$$

The difference between (2) and (3) is that in the latter, dependence between out-of-coalition features and those within the coalition is broken. In theory, *observing* $\boldsymbol{x}_{\mathcal{C},t}$ would change the distribution of the out-of-coalition features if the random variables were connected through latent effects. However, by *intervening* on a coalition, this distribution is unaffected. To illustrate this, consider two random variables, $X$ and $Y$, with the causal relationship in Figure 1.

Suppose we observe $X = x$, the observational conditional distribution describes: *the distribution of $Y$ given that $X$ is observed to take on the value $x$*, written as $p(y|x) = p(x, y)/p(x)$. The interventional conditional distribution describes instead: *the distribution of $Y$ given that we artificially set the value of $X$ to $x$*, denoted $p(y|do(x))$, obtained by assuming that $Y$ is dis-

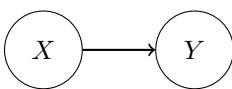

Figure 1: Causal graph indicating a direct effect between two random variables, $X$ and $Y$.

tributed by the original data generating process. Graphically, an intervention will remove all of the edges going into the corresponding variable. Consequently, we get that, $p(y|do(x)) = p(y|x)$ but $p(x|do(y)) = p(x)$. This means that the distribution of $y$ under the *intervention $X = x$* is equivalent to $y$ *conditioned* on $X = x$, yet for $Y = y$, $X$ and $Y$ become disconnected, hence $x$ has no effect on $y$, which is simply sampled from its marginal distribution.

**Computation**  These two lifts also differ in their relative computational expenditure (Lundberg & Lee, 2017). In particular, it is generally intractable to evaluate the conditional expectation of the loss function, requiring complex and costly methods for approximation (Covert et al., 2021). Conversely, cheap and simple methods exist to intervene on features (Sundararajan & Najmi, 2020). Whilst the most suitable method for evaluating the conditional expectation is disputed in literature (Chen et al., 2022), one such method requires training separate models for each subset of features; if each model is optimal with respect to the loss, then this is equivalent to marginalizing out features using their conditional distribution. In our linear regression setup, fitting a model for each coalition and evaluating the loss have running times of $\mathcal{O}(T \cdot |\mathcal{C}|^2 + |\mathcal{C}|^3)$ and $\mathcal{O}(T \cdot |\mathcal{C}|)$, respectively, hence whilst this approach is common, it scales poorly to high dimensions (Covert et al., 2021). In contrast, the interventional lift can be computed much faster by simply imputing out-of-coalition features, which requires training a single model (i.e., the grand coalition) so each permutation is computed in linear time. Note that, both lifts preserve the axioms of the original Shapley value, and subsequently the market properties provided, albeit in expectation. Furthermore, using Monte-Carlo estimation to approximate the Shapley values preserves the original form of the marginal contribution, $\Delta_i(\theta)$, for each permutation, thus effects of each lift generalize to arbitrary problem sizes.

**Causal Perspectives**  Observe that, if features are independent, both lifts are equivalent. Specifically, Janzing et al. (2020) showed that by distinguishing between the *true* features and those actually used as *input* to the model, we get that $p(\boldsymbol{x}_{\overline{\mathcal{C}},t} | do(\boldsymbol{x}_{\mathcal{C},t})) = p(\boldsymbol{x}_{\overline{\mathcal{C}},t})$. We can then calculate (3) from

(2) by simply replacing $p(\boldsymbol{x}_{\overline{\mathcal{C}},t}|\boldsymbol{x}_{\mathcal{C},t})$ with the marginal distribution, which would be equivalent when features are independent. With this in mind, we can use the following theorem to analyse these lifts from a causal perspective.

**Theorem 3.1.** *Marginal contributions derived using the observational conditional expectation formulation for $v(\cdot)$ as defined in (2) can be decomposed into indirect and direct causal effects.*

*Proof.* Following (2), the marginal contribution of the $i$-th feature for a single permutation $\theta \in \Theta$ derived using the observational lift can be written as

$$\Delta_i(\theta) = v^{\text{obs}}(\underline{\mathcal{C}}) - v^{\text{obs}}(\underline{\mathcal{C}} \cup i),$$

$$= \int \mathbb{E}\left[\ell_t|\boldsymbol{x}_{\underline{\mathcal{C}},t},\boldsymbol{x}_{\overline{\mathcal{C}}\cup i,t}\right] p(\boldsymbol{x}_{\overline{\mathcal{C}}\cup i,t}|\boldsymbol{x}_{\underline{\mathcal{C}},t})d\boldsymbol{x}_{\overline{\mathcal{C}}\cup i,t} - \int \mathbb{E}\left[\ell_t|\boldsymbol{x}_{\underline{\mathcal{C}}\cup i,t},\boldsymbol{x}_{\overline{\mathcal{C}},t}\right] p(\boldsymbol{x}_{\overline{\mathcal{C}},t}|\boldsymbol{x}_{\underline{\mathcal{C}}\cup i,t})d\boldsymbol{x}_{\overline{\mathcal{C}},t},$$

$$= \underbrace{\int \mathbb{E}\left[\ell_t|\boldsymbol{x}_{\underline{\mathcal{C}},t},\boldsymbol{x}_{\overline{\mathcal{C}}\cup i,t}\right] p(\boldsymbol{x}_{\overline{\mathcal{C}}\cup i,t}|\boldsymbol{x}_{\underline{\mathcal{C}},t})d\boldsymbol{x}_{\overline{\mathcal{C}}\cup i,t} - \int \mathbb{E}\left[\ell_t|\boldsymbol{x}_{\underline{\mathcal{C}}\cup i,t},\boldsymbol{x}_{\overline{\mathcal{C}},t}\right] p(\boldsymbol{x}_{\overline{\mathcal{C}},t}|\boldsymbol{x}_{\underline{\mathcal{C}},t})d\boldsymbol{x}_{\overline{\mathcal{C}},t}}_{\text{Direct effect}}$$

$$\underbrace{+ \int \mathbb{E}\left[\ell_t|\boldsymbol{x}_{\underline{\mathcal{C}}\cup i,t},\boldsymbol{x}_{\overline{\mathcal{C}},t}\right] p(\boldsymbol{x}_{\overline{\mathcal{C}},t}|\boldsymbol{x}_{\underline{\mathcal{C}},t})d\boldsymbol{x}_{\overline{\mathcal{C}},t} - \int \mathbb{E}\left[\ell_t|\boldsymbol{x}_{\underline{\mathcal{C}}\cup i,t},\boldsymbol{x}_{\overline{\mathcal{C}},t}\right] p(\boldsymbol{x}_{\overline{\mathcal{C}},t}|\boldsymbol{x}_{\underline{\mathcal{C}}\cup i,t})d\boldsymbol{x}_{\overline{\mathcal{C}},t}}_{\text{Indirect effect}},$$

where $\underline{\mathcal{C}} = \{j : j \prec_\theta i\}$ and $\overline{\mathcal{C}} = \{j : j \succ_\theta i\}$. This is the difference in the loss function when: (i) the value of the $i$-th feature is observed and the distribution of the remaining out-of-coalition features is unchanged (i.e., direct effect) and (ii) the distribution of the other out-of-coalition features does changed as a result of observing the $i$-th feature (i.e., indirect effect). $\qquad\square$

Following Theorem 3.1, we can see that by replacing conditioning by observation with the marginal distribution as in (2), the indirect effect expression disappears entirely. Hence, using the interventional lift removes consideration of causal effects *between features*, and subsequently any root causes with *indirect effects* (Heskes et al., 2020). As a result, the interventional lift is more effective at crediting features upon which the regression model has an explicit algebraic dependence. In contrast, the observational lift attributes features in proportion to indirect effects (Frye et al., 2020b), which some argue is illogical as features not explicitly used by the model have the possibility of receiving non-zero allocation.

Whilst this dispute has been used to reject the general use of Shapley values for model interoperability in machine learning (Kumar et al., 2020) and argue that Lundberg & Lee (2017) were mistaken to only convey (3) as a cheap approximation of (2) (Janzing et al., 2020), the choice between the observational and interventional lifts can in fact be viewed as conditional on whether one intends to be *true to the data* or *true to the model*, respectively, meaning the trade-offs of each approach can be seen as context-specific (Chen et al., 2020).

**Interpreting Rewards**   We can explore this last conjecture by considering how the rewards of the support agents may differ depending on the choice of lift. We know that the predictive performance of the regression model out-of-sample is contingent upon the availability of features that were used during training, which, in practice, requires data of the support agents to be streamed continuously in a timely fashion, particularly for an online setup. If a feature was missing, the efficacy of the forecast may drop, the extent to which would relate not to any root causes or indirect effects regarding the data generating process, but rather the magnitude of direct effects.

Specifically, larger rewards would be made to support agents with features to which the predictive performance of the model is most sensitive, providing incentives to reduce data being unavailability, resembling reserve payments in electricity markets, where assets are remunerated for being available in times of need. For the observational lift, it would instead be unclear as to whether comparatively larger rewards in the regression market are consequential of features having a sizeable impact on predictive performance, or merely a result of indirect effects through those that do. The interventional lift therefore better aligns with desirable intentions of the market.

**Risk Implications**   When features are strongly correlated, conditioning by intervention can lead to model evaluation on points outwith the true data manifold (Frye et al., 2020a). This can visualized

with the simple illustration in Figure 2. Intervening on independent features yields samples within the original data manifold. However, when this is not the case, there is a possibility for extrapolating far beyond the training distribution, where model behaviour is unknown. In the remainder of this section we consider what impact this may have on the market outcomes.

Multicollinearity inflates the variance of the coefficients, which can distort the estimated mean when the number of in-sample observations is limited. The posterior variance of the $i$-th coefficient can be written as $var(w_i) = \kappa_i/\xi|\mathcal{D}_t|$, where $\xi$ is the intrinsic noise precision of the target and $\kappa_i$ is the variance inflation factor, given by $\kappa_i = \mathbf{e}_i^\top(\sum_{t\leq t} \boldsymbol{x}_t^\top \boldsymbol{x}_t)^{-1}\mathbf{e}_i$, $\forall i \in \mathcal{I}$, where $\mathbf{e}_i$ is the $i$-th basis vector. Whilst $\kappa_i \geq 1$, it has no upper bound, meaning $\kappa_i \mapsto \infty$, $\forall i$, with increasing collinearity.

From a variance decomposition perspective, the expected Shapley value of the $i$-th feature can be shown to be equivalent to the variance in the target signal that it explains, such that, $\mathbb{E}[\phi_i] = \mathbb{E}[w_i]^2\, var(X_i)$, approximating the behaviour of the interventional Shapley value when features are correlated (Owen & Prieur, 2017).

As the posterior is Gaussian, the Shapley values follow a noncentral Chi-squared distribution with one degree of freedom. We can write the PDF of the distribution of the Shapley value in closed-form as $p(\phi_i)/(var(X_i)var(w_i)) = \sum_{k=0}^{\infty}(1/k!)e^{\eta/2}(\eta/2)^k)\chi^2(1+2k)$, $\forall i$, where the noncentral Chi-squared distribution is seen to simply be given by a Poisson-weighted mixture of central Chi-squared distributions, $\chi^2(\cdot)$, with noncentrality $\eta = \mathbb{E}[w_i]^2/var(w_i)$, for which the moment generating function is known in closed form.

By deriving the second moment, $\frac{1}{2}var(\phi_i) = var(w_i)\left(2\mathbb{E}[w_i]^2 + var(w_i)\right)\left(var(X_i)\right)^2$, $\forall i$, we see that the variance of the attribution for any feature is a quadratic function of the vari-

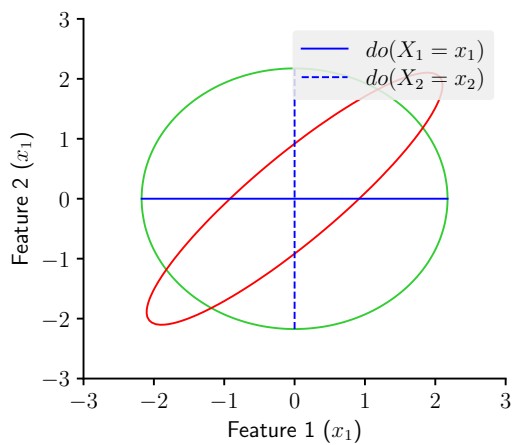

Figure 2: Interventions producing points outwith the data manifold. Green and red lines are level sets denoting the 0.99 quantile of the training data when features are independent and correlated.

ance of the corresponding coefficient, thus the variance inflation induced by multicollinearity. Nevertheless, this problem vanishes with increasing sample size, as $var(w_i) \mapsto 0$, $\forall i$ (Qazaz et al., 1997). If a limited number of observations are available, distorted revenues could be remedied using *zero-Shapley* or *absolute-Shapley* proposed in Liu (2020), or restricting evaluations to the data manifold (Taufiq et al., 2023).

## 4   ROBUSTNESS TO REPLICATION

Although it is natural for datasets to contain some overlapping information, in our analytics market such redundancy may also arise as a result of replication. The fact that data can be freely replicated differentiates it from material commodities—a motive for reassessing fundamental mechanism design concepts (Aiello et al., 2001). For example, a simple second price auction becomes impractical unless sellers somehow limit the number of replications, which may in turn curtail revenue. In this section, we demonstrate how the observational lift provides incentives for replication, the downsides of this, and how these can be remedied by instead adopting the interventional lift.

**Definition 4.1.** *A replicate of the $i$-th feature is defined as $x'_{i,t} = x_{i,t} + \eta$, where $\eta$ represents centred noise with finite variance, conditionally independent of the target given the feature.*

Under Definition 4.1, the observational lift described in (2) provides a monetary incentive for support agents to replicate their data and act under multiple (false) identities. To illustrate this, consider the causal graph in Figure 3. Suppose that $x_{1,t}$ and $x_{2,t}$ are identical features, such that $w_1 = w_2$, and that each is owned by a unique support agent, $a_1$ and $a_2$, respectively. With Theorem 3.1, the reward to each support agent before any replication is made will be $\pi/2$, where recall $\pi$ is the total market

revenue. Now suppose that $a_2$ replicates their feature $k$ times and for ease assume $var(\eta) = 0$. Using the same logic, the revenues of agents $a_1$ and $a_2$ will be $\pi/(2+k)$ and $\sum_{1+k} \pi/(2+k) = \pi(1+k)/(2+k)$, respectively. Hence a malicious agent can simply replicate their data many times so as to maximize their overall revenue, and diminish that of others.

**Definition 4.2.** *Let $\boldsymbol{x}_t^+$ denote the original feature vector augmented to include any additional replicates, with an analogous index set, $\mathcal{I}^+$. According to Agarwal et al. (2019), a market is replication-robust if $\pi_a^+ \le \pi_a$, $\forall a \in \mathcal{A}_{-c}$, where $\pi_a^+$ is the new revenue derived using $\boldsymbol{x}_t^+$ instead.*

In attempt to remedy this issue, the authors in Agarwal et al. (2019) propose *Robust-Shapley*, $\phi_i^{\text{rob}} = \phi_i \exp(-\gamma \sum_j s(X_{i,t}, X_{j,t}))$, where $s(\cdot, \cdot)$ is a similarity metric (e.g., cosine similarity). This method penalizes similar features so as to remove the incentive for replication, satisfying Definition 4.2. However, this means that not only replicated features are penalized, but also those with naturally occurring correlations between features. As a result, budget balance is lost, the extent to which depends on the chosen similarly metric and the value of $\gamma$.

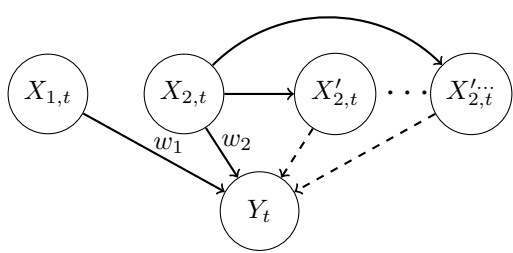

Figure 3: Direct effects (solid) and indirect effects (dashed) induced by replicating $X_{2,t}$. The prime superscript denotes a replicated feature.

A similar result is presented in Han et al. (2023) who consider the general set of semivalues, the class of solution concepts to submodular games to which the Shapley value belongs (Dubey et al., 1981). The authors show that the way in which a semivalue weights coalition sizes has an affect on the resultant properties, and that the Banzhaf value (Lehrer, 1988) is in fact replication-robust by design (i.e., with respect to Definition 4.2), along with many other semivalues, albeit still penalizing naturally occurring correlations. That being said, Definition 4.2 leaves the market susceptible to spiteful agents—those willing to sacrifice their revenue in order to minimize that of others. As such, we refer to this definition as *weakly* robust.

**Proposition 4.3.** *A Shapley-value based attribution policy based on the interventional lift instead yields a stricter notion of being replication-robust, such that $\pi_a^+ \equiv \pi_a$, $\forall a \in \mathcal{A}_{-c}$.*

*Proof.* With Definition 4.1, each replicate in $\boldsymbol{x}_t^+$ only induces an indirect effect on the target. However, from Theorem 3.1, we know that the interventional lift only captures direct effects. Therefore, for each of the replicates, we write the marginal contribution for a single permutation $\theta \in \Theta$ as

$$\Delta_i(\theta) = v^{\text{int}}(\mathcal{C}) - v^{\text{int}}(\mathcal{C} \cup i),$$

$$= \int \mathbb{E}\left[\ell_t | \boldsymbol{x}_{\mathcal{C},t}, \boldsymbol{x}_{\overline{\mathcal{C}} \cup i, t}\right] p(\boldsymbol{x}_{\overline{\mathcal{C}} \cup i, t} | \boldsymbol{x}_{\mathcal{C},t}) d\boldsymbol{x}_{\overline{\mathcal{C}} \cup i, t} - \int \mathbb{E}\left[\ell_t | \boldsymbol{x}_{\mathcal{C} \cup i, t}, \boldsymbol{x}_{\overline{\mathcal{C}},t}\right] p(\boldsymbol{x}_{\overline{\mathcal{C}},t} | \boldsymbol{x}_{\mathcal{C},t}) d\boldsymbol{x}_{\overline{\mathcal{C}},t},$$

$$= 0, \quad \forall i \in \mathcal{I}_{-c}^+ \setminus \mathcal{I}_{-c},$$

and therefore $\phi_i \propto \sum_{\theta \in \Theta} \Delta_i(\theta) = 0$ for each of the replicates. For the original features, any direct effects will remain unchanged, as visualized in Figure 3. This leads to

$$\pi_a^+ = \sum_{i \in \mathcal{I}_a} \lambda \mathbb{E}[\phi_i] + \sum_{i \in \mathcal{I}_a^+ \setminus \mathcal{I}_a} \lambda \underbrace{\mathbb{E}[\phi_i]}_{=0} = \pi_a, \quad \forall a \in \mathcal{A}_{-c},$$

showing that by replacing the conventional observational lift with the interventional lift, the Shapley value-based attribution policy is *strictly* robust to both replication *and* spitefulness by design. $\qquad\square$

# 5 EXPERIMENTAL ANALYSIS

We now validate our key findings on a real-world case study. We use an open source dataset to aid reproduction of our work, namely the Wind Integration National Dataset (WIND) Toolkit, detailed in Draxl et al. (2015). Our setup is a stylised electricity market setup where agents—in our case, wind producers—are required to notify the system operator of their expected electricity generation in a forward stage, one hour ahead of delivery, for which they receive a fixed price per unit. In real-time, they receive a penalty for deviations from the scheduled production, thus their downstream revenue is an explicit function of forecast accuracy.

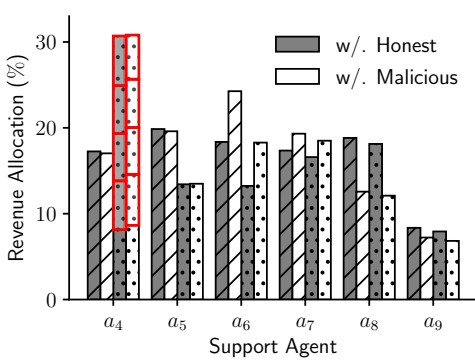 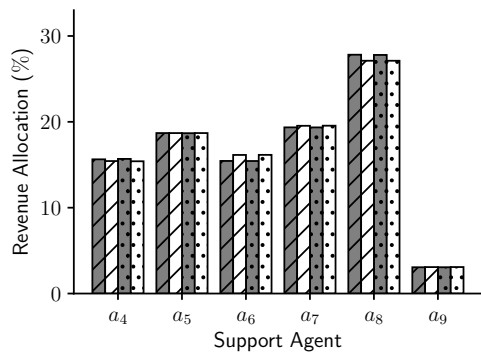

(a) *Observational*: Revenue of $a_4$ is increased due to indirect effects induced by the replicates.

(b) *Interventional*: Revenue of $a_4$ remains the same by accounting only for direct effects.

Figure 4: Revenue allocations for each support agent for both (a) observational and (b) interventional lifts, when agent $a_4$ is honest ($//$) and malicious ($\circ$), by replicating their feature. The gray and white bars correspond to in-sample and out-of-sample market stages, respectively. The revenue split amongst replicates is depicted by the stacked bars highlighted in red.

**Methodology**   The dataset contains wind power measurements simulated for 9 wind farms in South Carolina, all located within 150 km of each other. Whilst this data is not exactly *real*, it captures the spatio-temporal aspects of wind power production, with the benefit of remaining free from any spurious records, as can often be the case with real-world datasets. For simplicity, we let $a_1$ be the central agent, however each could assume this role in parallel.

We use the regression framework described in Section 2, with an *Auto-Regressive with eXogenous input* model, such that each agent is assumed to own a single feature, namely a 1-hour lag of their power measurement. We focus on assessing rewards rather than competing with state-of-the-art forecasting methods, so we use a very short-term lead time. Nevertheless, our mechanism readily allows more complex models for those aiming to capture specific intricacies of wind power production, for instance the bounded extremities of the power curve (Pinson, 2012).

We perform a pre-screening, such that given the redundancy between the lagged measurements of $a_2$ and $a_3$ with that of $a_1$, we remove them from the market in line with our assumptions. The first 50% of observations are used to clear the in-sample regression market and fit the regression model, whilst the latter half are used for the out-of-sample market. We clear both markets considering each agent is honest, that is, they each provide a single report of their true data. Next, we re-clear the markets, but this time assume agent $a_4$ is malicious, replicating their data, thereby submitting multiple separate features to the market to increase their revenue. This problem size doesn't require approximate Shapley values, but recall findings hold either way, and generalize theoretically to arbitrary numbers of agents. Each market clearing was solved on CPU hardware (Intel Xeon E5-2686 v4, 2.3 GHz)

**Results**   We start by setting the number of replicates $k = 4$, with valuation $\lambda = 0.5$ USD per time step and per unit improvement in $\ell$, for both in-sample and out-of-sample market stages. However, we are primarily interested in reward allocation rather than the magnitude—see Pinson et al. (2022) for a complete analysis of the monetary incentive to each agent participating in the market. Overall the in-sample and out-of-sample losses improved by 10.6% and 13.3% respectively with the help of the support agents. In Figure 4, we plot the allocation for each agent with and without the malicious behavior of agent $a_4$, for both lifts. When this agent is honest, we observe that the observational lift spreads credit relatively evenly amongst most features, suggesting that many of them have similar indirect effects on the target. The interventional lift favours agent $a_8$, which, as expected, owns the features with the greatest spatial correlation with the target. In this market, most of the additional revenue of agent $a_8$ appears to be lost from agent $a_9$ compared with the observational lift, suggesting that whilst these features are correlated, it is agent $a_8$ with the greatest direct effect.

When agent $a_4$ replicates their data, with the observational lift, agents $a_5$ to $a_8$ earn less, whilst agent $a_4$ earns more. This shows that this lift indeed spreads rewards proportionally amongst indirect

effects, of which there are four more due to the replicates, and so the malicious agent out-earns the others. Since the interventional lift only attributes direct effects, each replicate gets zero reward, so the malicious agent is no better off than before. Rewards were consistent between in-sample and out-of-sample, likely due to the large batch size and limited nonstationarities within the data.

To compare our work against current literature, in Figure 5 we plot the allocation of agent $a_4$ with increasing number of replicates. Here, *Robust-Shapley* and *Banzahf Value* refer to both the penalization approach of Agarwal et al. (2019) and the use of another semivalue in Han et al. (2023), respectively. With the observational lift, the proportion of revenue obtained increases with the number of replicates, as in the previous experiment. With *Robust-Shapley*, the allocation indeed decreases with the number of replicates, demonstrating this approach is *weakly* replication-robust, but is considerably less compared with the other approaches since natural similarities are also penalized. The authors argue this is an incentive for provision of unique information, but this allows agents to be spiteful. The *Banzahf Value* is strictly robust to

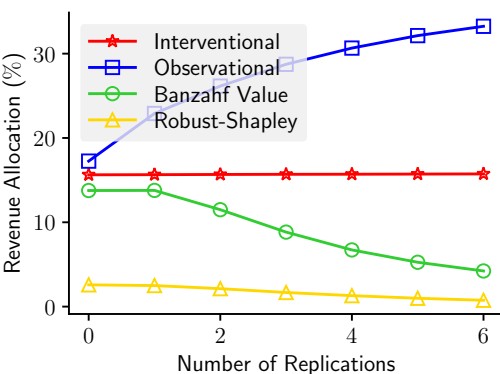

Figure 5: Revenue allocation of agent $a_4$.

replicaiton for $k = 0$, but only weakly for $k \geq 1$. Lastly, unlike these methods, our proposed use of the interventional lift remains strictly replication-robust throughout as expected, with agent $a_4$ not able to benefit from replicating their feature, without penalizing the other agents.

## 6 CONCLUSIONS

Many machine learning tasks could benefit from using the data of others, however convincing firms to share information, even if privacy is assured, poses a considerable challenge. Rather than relying on data altruism, data markets are recognized as a promising way of providing incentives for data sharing, many of which use Shapley values to allocate rewards. Nevertheless, there are a number of open issues that remain before such mechanisms can be used in practice, one of which is vulnerability to replication incentives, which we showed leads to undesirable reward allocation and restricts the practical viability of these markets.

We introduced a general framework for data markets for supervised learning problems that subsumes many of these existing proposals. We demonstrated that there are several different ways to formulate a machine learning task as cooperative game and analysed their differences from a causal perspectives. We showed that use of the observational lift to value a coalition is the source of these replication incentives, which many works have tried to remedy through penalization methods, which facilitate only *weak* robustness. Our main contribution is an alternative algorithm for allocating rewards that instead uses interventional conditional probabilities. Our proposal is robust to replication without comprising market properties such as budget balance. This is a step towards making Shapley value-based data markets feasible in practice.

From a causal perspective, the interventional lift has additional potential benefits, including reward allocations that better represent the reliance of the model on each feature, providing an incentive for timely and reliable data streams for useful features, that is, those with greater influence on predictive performance. It is also favourable with respect to computational expenditure. There is of course, no free lunch, as using the interventional conditional expectation can yield undesirable rewards when feautres are highly correlated and the number of observations is low. Nevertheless, future work could examine the extent to which the mentioned remedies mitigate this issue, as well as their impact on the market outcomes. Ultimately, when it comes to data valuation, the Shapley value is not without its limitations—it is not generally well-defined in a machine learning context and requires strict assumptions, not to mention its computational complexity. This should also incite future work into alternative mechanism design frameworks, for example those based on non-cooperative game theory instead.

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
