# OpenReview forum: "Towards Replication-Robust Data Markets"
_ICLR.cc/2025/Conference — ICLR 2025 Conference Withdrawn Submission_

### Official Review · Reviewer_7bxy · 2024-10-31

**Soundness:** 2
**Presentation:** 2
**Contribution:** 2
**Rating:** 3
**Confidence:** 3

**Summary:**

Shapley values are a popular way to attribute effort/contributions in classification tasks, and is popular in data market scenarios. One shortcoming they suffer from, however, is that they are not robust to replication. That is, an agent can increase their "contribution" by making copies of their features. This paper attempts to address this replication issue taking a causal perspective which differentiates between the direct and indirect contributions to the predictor and ignores the indirect aspect. This in turn has the effect of addressing the replicability issue that initiates the work.

In general, I found the paper quite difficult to read and parse. A lot of the notation and assumptions were stated in a very matter-of-fact way, with justification or implications being not fully addressed. There are also many instances where the paper hand-waves key aspects - for example, it mentions the computational challenges of Shapley Value and the benefits of using the interventional approach - but does not give any precise results. The work to me is largely theoretical in nature, but there are few strong theoretical results. Further, a causal perspective of Shapley Values is not novel and the authors do cite the (Heskes et al. 2020); however, I don't see the authors strongly arguing for their novel contributions on top of this. So while the observations of the paper are somewhat intriguing, I don't see the contribution strong enough to warrant a strong accept.

**Strengths:**

* The authors do highlight a major shortcoming of Shapley values in that it is vulnerable to replication attacks. This is especially prescient in data markets given that replication is essentially free.

* The causal approach proposed is an interesting way to mitigate this issue. It has been proposed for understanding feature importance in ML tasks (Heskes et al, 2020), and it is nice to see that it also addresses the replication attack issue.

**Weaknesses:**

* The writing and overall flow of the paper is not easy to follow. Many justifications or results are given in a very informal hand-wavy manner. I give some examples below:
    * In the model, a time index $t$ is mentioned but not clear why or what it represents. As far as I see, this isn't a dynamic problem and it's not really used elsewhere.
    * The authors give the result for a stylized regression problem with gaussian noise, where everything can be computed in closed form. They however mention this is only for illustrative purposes, but fail to substantiate this. For example, in line 132, they mention "a Gaussian framework is merely for mathematical convinience, and our framework can be readily extended to more general hypotheses."
    * It would help to explicitly give the algorithm for the computation of their modified Shapley value and a formal statement about computation.

* In (Agarwal et. al, 2019) they propose a robust Shapley Value where agents don't gain by a chosen $\varepsilon$ if they choose to replicate. I don't see any natural shortcomings of this approach - the authors here mention it fails budget balance but I'm not sure what exactly they mean by that. Why do we need to ensure revenue pre-replication is exactly equal to revenue post replication? Given we want to prevent replication, we simply want to ensure they don't gain by replicating.

* What is exact additional contribution here as compared to (Heskes et al, 2020) which also proposes a causal version of Shapley Values.  Is it mainly the observation that causal Shapley value addresses the replication issue? If so, I am not sure if it is significant enough given the many shortcomings of Shapley values - it's not afterall a widely deployed algorithm in this setting.

* There are several drawbacks to the interventional approach mentioned here. Mainly, it takes a very simplistic view of causal relations in model. For example, two features might indirectly influence each other through a latent variable. Am I correct to say that the proposed approach will ignore this relationship and thus undervalue one or both features. This in itself isn't a major criticism - all approaches have shortcomings. But I think it does warrant a more than one sentence discussion and should perhaps include the settings where this is more or less appropriate as a solution.

**Questions:**

See above

---

### Official Review · Reviewer_gcDi · 2024-11-01

**Soundness:** 3
**Presentation:** 2
**Contribution:** 2
**Rating:** 5
**Confidence:** 3

**Summary:**

The authors propose the use of Shapley values from a causal perspective (interventional conditional expectation) to determine the value of features to allocate rewards (revenue) in a marketplace.

**Strengths:**

- The authors clearly define the problem and the paper is generally well-written and easy to read.
- The authors compare the proposed method with various other methods.

**Weaknesses:**

**Weaknesses and questions**
- The introduction, in my opinion, does not sufficiently place authors’ work in the literature or clearly define the authors’ contributions.
- Given the use case, i.e., revenue allocation, I think it’s important to ensure stability and consistency of the valuation method. The proposed method has been shown to have several issues, especially the heavy reliance on the model (see, for example, Kumar et al. 2020). I am uncertain if the proposed method (Shapley feature attribution) would be the best for the use case (feature pricing).
- The experimental setup doesn’t sufficiently support the setup. For example, it would have been better to see more clear and intuitive cases of varied feature sources, for the same rows, and what y means in a more clearer sense. I understand it might be hard to find an already collected/cleaned dataset to experiment on. The authors could potentially collect various datasets and create a semi-synthetic dataset, or test with fully synthetic setups.
- I like the figure authors show for the performance of the proposed method specific to a4. I do however wonder if there are differences in how the different methods do specific to the utility function, because even though feature value (reward) might improve, the utility might not, which inadvertently influences the buyer's decision on which method to use.
- How well does the proposed model do in cases where several agents/sellers replicate data differently (with different methods on varied features, etc) and replication detection is beyond correlation with other features?


**Miscellaneous - did not necessarily affect my score**
- I think that "of" is missing in this sentence in the abstract: “to improve the predictions others”.
- There were several instances where the word features were misspelled as “feautres”, and replication as “replicaiton”. Additionally, when printed, figure 4a comes out as a dark figure with bars unclear/not seen.
- I think that compensating data contributors is an important problem. However, I think there are very instances where data owners have different feature information on the same agents, and those that do are likely to be organizations that are either restricted by privacy laws from sharing/selling that data or proprietary and likely to prefer techniques like federated learning over data selling.
- Authors do not provide their code and data.
- In my opinion, comparing federated learning with data markets through the lens of altruism is very limiting and doesn’t sufficiently cover the comparisons of the methods. For example, issues related to privacy/laws, proprietary data, computational expenses, etc, could offer better comparisons.

**Questions:**

The questions below mainly reflect the weaknesses I added, so authors should read the weakness section for a detailed explanation.
- Given the nature of the use case of the proposed method,  does the method ensure consistency and stability of the values given to different features?
- The experimental setup does not sufficiently support the method.
- In addition to robustness to replication, is there a difference in how well methods do on the utility function?  For the data buyer, is there potentially a tradeoff between performance on the utility function and chosen valuation method computational expenses over robustness to replication? And if that's the case, would the proposed method likely preferred over others?
- How well does the model handle diverse replication, and other varied "malicious agent" behavior (e.g. potentially injecting adversarial data)?

---

### Official Review · Reviewer_prgM · 2024-11-04

**Soundness:** 2
**Presentation:** 1
**Contribution:** 2
**Rating:** 3
**Confidence:** 2

**Summary:**

* This paper introduces a causal framework for data-sharing markets that addresses vulnerabilities to replication attacks.
* In the model, there is a market with agents $\\mathcal{A}$. One agent is a central agent $c\\in\\mathcal{A}$ seeking to improve predictions, and the rest are support agents which may sell their data. There is a set $\\mathcal{I}$ of features and each agent owns a subset $\\mathcal{I}_a \\subseteq \\mathcal{I}$. The central agent makes predictions about a target signal $y$ according to a linear regression model. The model assumes a Gaussian prior for weights, and defines market revenue as the expected negative logarithm of the predictive density, multiplied by an exogenous parameter $\\lambda$. While value is ideally distributed according to Shapley values, the paper notes that calculating these values is complex and computationally demanding.
* To overcome the computational barrier with Shapley values, the authors propose a method based on Pearl’s do-calculus. Theorem 3.1 presents a decomposition of the marginal contributions into indirect and direct causal effects.
* Section 4 analyzes the model’s robustness to replication, showing that agents can exploit replication for increased revenue. The authors argue that previous solutions are susceptible to spiteful behaviors, and they demonstrate that interventional Shapley values ensure robustness against both replication and spitefulness (Proposition 4.3).
* Finally, Section 5 provides an empirical evaluation using simulated data inspired by wind-farm market dynamics, showing favorable results.

**Strengths:**

* The paper addresses a well-motivated problem.
* Frequent references to related work provides context and highlights the limitations of existing methods.

**Weaknesses:**

* The progression of ideas could be made clearer to enhance readability.
* The applicability of the proposed approach is unclear. Section 6 mentions that “using the interventional conditional expectation can yield undesirable rewards when features are highly correlated and the number of observations is low”, but this statement lacks an explanation.
* No code is provided, which limits verification and further research.
* Minor issues:
  * In eq. (1), the notation $i\\in |\\mathcal{I}_c|$ is confusing, as $|\\mathcal{I}_c|$ usually denotes set size, a scalar.
  * Typo in L480: “feautres.”

**Questions:**

* Would it be possible to provide a practical illustrative example of the model, utilizing the key definitions and demonstrating the utility of the causal approach?
* What is the role of time $t$ in the analysis?
* Would it be possible to explain the experimental setup in more detail and share the code used for the analysis?
* In L136: What motivates the choice of the negative log likelihood for model fitting?
* In L130: What is the definition of $w_\\mathcal{C}$?

---

### Official Review · Reviewer_vx1s · 2024-11-05

**Soundness:** 3
**Presentation:** 1
**Contribution:** 2
**Rating:** 5
**Confidence:** 2

**Summary:**

The paper proposes a replication-robust data valuation method for linear models. They use Pearl’s do-calculus from causal reasoning to derive Shapley value-based rewards that are robust to this malicious replication, whilst preserving desirable market properties. They also show the effectiveness of their approach by simulations.

**Strengths:**

The paper proposes a novel data valuation approach that is robust to data replication, based on the commonly used Shapley value. The application of Pearl’s do-calculus is interesting and novel.

**Weaknesses:**

- While the proposed method has the nice property of replication-robustness, it is unclear whether the calculated data value effectively predicts the data's usefulness in practice. For instance, does high-value data lead to models with better performance? Additionally, could the use of interventional lift harm the prediction of a data point's contribution in practice?

- The paper assumes a Bayesian model, which may limit its use in machine learning, where frequentist approaches are more prevalent. It is not immediately clear how this method can be adapted for general supervised machine learning algorithms.

- Although the idea of using interventional lift is innovative, many of the technical results appear to be drawn directly from previous work, such as the computational results.

**Minor comments:**

The paper assumes substantial prior knowledge, which may not be accessible to all readers.
- The Market Clearing section in the Preliminaries is unclear. Even with some background in game theory, I was unable to grasp the concept of a two-stage batch market after reading this section.
- Additionally, the paper presumes familiarity with causality; without a background in this area, I found Section 3 extremely difficult to follow.
- "Hence, the characteristic function must lift the original loss to simulate removal of features" What does **lift** mean here?

There are several closely related paper not cited. [1] proposes a replication robust data valuation method for linear models. The method proposed by [2] is also robust to replication in the same Bayesian setting.
- [1] Xu et al., 2021. Validation Free and Replication Robust Volume-based Data Valuation
- [2] Chen et al., 2020. Truthful Data Acquisition via Peer Prediction

**Questions:**

- Does the high value data give a model with better performance? Will the use of interventional lift harm the prediction of data point's contribution in practice?
- What is the purpose of the paragraphs under "Causal Perspectives"?

---

### Note · Authors · 2024-11-13

I have read and agree with the venue's withdrawal policy on behalf of myself and my co-authors.